# CAdam: Confidence-Based Optimization for Online Learning

## Abstract

Modern recommendation systems frequently employ online learning to dynamically update their models with freshly collected data. The most commonly used optimizer for updating neural networks in these contexts is the Adam optimizer, which integrates momentum ($m_t$) and adaptive learning rate ($v_t$). However, the volatile nature of online learning data, characterized by its frequent distribution shifts and presence of noises, poses significant challenges to Adam's standard optimization process: (1) Adam may use outdated momentum and the average of squared gradients, resulting in slower adaptation to distribution changes, and (2) Adam's performance is adversely affected by data noise. To mitigate these issues, we introduce CAdam, a confidence-based optimization strategy that assesses the consistence between the momentum and the gradient for each parameter dimension before deciding on updates. If momentum and gradient are in sync, CAdam proceeds with parameter updates according to Adam's original formulation; if not, it temporarily withholds updates and monitors potential shifts in data distribution in subsequent iterations. This method allows CAdam to distinguish between the true distributional shifts and mere noise, and adapt more quickly to new data distributions. Our experiments with both synthetic and real-world datasets demonstrate that CAdam surpasses other well-known optimizers, including the original Adam, in efficiency and noise robustness. Furthermore, in large-scale A/B testing within a live recommendation system, CAdam significantly enhances model performance compared to Adam, leading to substantial increases in the system's gross merchandise volume (GMV).

## 1 Introduction

Modern recommendation systems, such as those used in online advertising platforms, rely on online learning to update real-time models with freshly collected data batches (Ko et al., 2022). In online learning, models continuously adapt to users' interests and preferences based on immediate user interactions like clicks or conversions. Unlike traditional offline training—where data is pre-collected and static—online learning deals with streaming data that is often noisy and subject to frequent distribution changes. This streaming nature makes it challenging to effectively denoise and reorganize training samples (Su et al., 2024; Zhang et al., 2021).

A widely adopted optimizer in these systems is the Adam optimizer (Kingma & Ba, 2015), which combines the strengths of parameter-adaptive methods and momentum-based methods. Adam adjusts learning rates based on the averaged gradient square norm ($v_t$) and incorporates momentum ($m_t$) for faster convergence. Its ability to maintain stable and efficient convergence by dynamically adjusting learning rates based on the first and second moments of gradients has made it a reliable choice for optimizing deep learning models across diverse applications, including image recognition (Alexey, 2020), natural language processing (Vaswani, 2017), and reinforcement learning (Schulman et al., 2017). However, Adam faces significant challenges in online learning environments. Specifically, it treats all incoming data equally, regardless of whether it originates from the original distribution, a new one, or is merely noise. This indiscriminate treatment leads to two key problems:

1. **Outdated Momentum and Averaged Squared Gradients**: When the data distribution shifts—a common occurrence in online systems due to factors such as daily cycles in shopping habits, rapidly changing trends on social media, seasonal changes, promotional events,

and sudden market dynamics—Adam continues to use momentum and averaged squared gradients computed from previous data (Lu et al., 2018; Viniski et al., 2021). These outdated statistics can misguide the optimizer, resulting in slower adaptation to the new data distributions.

2. **Sensitivity to Noise**: Online learning data often contains noisy labels (Yang et al., 2023). For example, in advertisement systems, users might click ads by mistake (false positives) or ignore ads they are interested in (false negatives) (Wang et al., 2021). Sensitivity to such noise can affect convergence speed and may cause parameters to deviate from the correct optimization direction, especially in scenarios where noisy data constitutes a large proportion.

To address these issues inherent in online learning with Adam, we propose Confidence Adaptive Moment Estimation (CAdam), a novel optimization strategy that enhances Adam's robustness and adaptability. CAdam introduces a confidence metric that evaluates whether updating a specific parameter will be beneficial for the system. This metric is calculated by assessing the alignment between the current momentum and the gradient for each parameter dimension.

Specifically, if the momentum and the gradient point in the same direction, indicating consistency in the optimization path, CAdam proceeds with the parameter update following Adam's rule. Otherwise, if they point in opposite directions, CAdam pauses the update for that parameter to observe potential distribution changes in subsequent iterations. This strategy hinges on the idea that persistent opposite gradients suggest a distributional shift, as the momentum (an exponential moving average of past gradients) represents the recent trend. If the opposite gradients do not persist, it it likely to be noise, and the model resumes normal updates, effectively filtering out the noise.

By incorporating this simple, plug-and-play mechanism, CAdam retains the advantages of momentum-based optimization while enhancing robustness to noise and improving adaptability to meaningful distribution changes in online learning scenarios.

Our contribution can be summarized as follows:

1. We introduce CAdam, a confidence-based optimization algorithm that improves upon the standard Adam optimizer by addressing its limitations in handling noisy data and adapting to distribution shifts in real-time online learning.

2. Through extensive experiments on both synthetic and public datasets, we demonstrate that CAdam consistently outperforms popular optimizers in online recommendation settings.

3. We validate the real-world applicability of CAdam by conducting large-scale online A/B tests in a live system, proving its effectiveness in boosting system performance and achieving significant improvements in gross merchandise volume (GMV) worth millions of dollars.

## 2 RELATED WORK

**Adam Extensions** Adam is one of the most widely used optimizers, and researchers have proposed various modifications to address its limitations. AMSGrad (Reddi et al., 2018) addresses Adam's non-convergence issue by introducing a maximum operation in the denominator of the update rule. RAdam (Liu et al., 2019) incorporates a rectification term to reduce the variance caused by adaptive learning rates in the early stages of training, effectively combining the benefits of both adaptive and non-adaptive methods. AdamW (Loshchilov, 2017) separates weight decay from the gradient update, improving regularization. Yogi (Zaheer et al., 2018) modifies the learning rate using a different update rule for the second moment to enhance stability. AdaBelief (Zhuang et al., 2020) refines the second-moment estimation by focusing on the deviation of the gradient from its exponential moving average rather than the squared gradient. This allows the step size to adapt based on the "belief" in the current gradient direction, resulting in faster convergence and improved generalization. Our method, CAdam, similarly leverages the consistency between the gradient and momentum for adjustments. However, it preserves the original update structure of Adam and considers the sign (directional consistency) between momentum and gradient, rather than their value deviation, leading to better performance under distribution shifts and in noisy environments.

**Adapting to Distributional Changes in Online Learning** In online learning scenarios, models encounter data streams where the underlying distribution can shift over time, a phenomenon known as concept drift (Lu et al., 2018). Adapting to these changes is essential for maintaining model performance. One common strategy is to use sliding windows or forgetting mechanisms (Bifet & Gavalda, 2007), which focus updates on the most recent data. Ensemble methods (Street & Kim, 2001) maintains a collection of models trained on different time segments and combine their predictions to adapt to emerging patterns. Adaptive learning algorithms, such as Online Gradient Descent (Zinkevich, 2003), dynamically adjust the learning rate or model parameters based on environmental feedback. Meta-learning approaches (Finn et al., 2017) aim to develop models that can quickly adapt to new tasks or distributions with minimal updates. Additionally, (Viniski et al., 2021) demonstrated that streaming-based recommender systems outperform batch methods in supermarket data, particularly in handling concept drifts and cold start scenarios.

**Robustness to Noisy Data** General methods for noise robustness include robust loss functions (Ghosh et al., 2017), which modify the objective function to reduce sensitivity to mislabeled or corrupted data; regularization techniques (Srivastava et al., 2014), which prevent overfitting by introducing noise during training; and noise-aware algorithms (Gutmann & Hyvärinen, 2010), which explicitly model noise distributions to improve learning. In recommendation systems, enhancing robustness against noisy data is crucial and is typically addressed through two main strategies: *detect and correct* and *detect and remove*. *Detect and correct* methods, such as AutoDenoise (Ge et al., 2023) and Dual Training Error-based Correction (DTEC) (Panagiotakis et al., 2021), identify noisy inputs and adjust them to improve model accuracy by leveraging mechanisms like validation sets or dual error perspectives. Conversely, *detect and remove* approaches eliminate unreliable data using techniques such as outlier detection with statistical models (Xu et al., 2022) or semantic coherence assessments (Saia et al., 2016) to cleanse user profiles. While these strategies can effectively enhance recommendation quality, they often require explicit design and customization for specific models or tasks, limiting their general applicability.

## 3 DETAILS OF CADAM OPTIMIZER

**Notations** We use the following notations for the CAdam optimizer:

- $f(\theta) \in \mathbb{R}, \theta \in \mathbb{R}^d$: $f$ is the stochastic objective function to minimize, where $\theta$ is the parameter vector in $\mathbb{R}^d$.
- $g_t$: the gradient at step $t$, $g_t = \nabla_\theta f_t(\theta_{t-1})$.
- $m_t$: exponential moving average (EMA) of $g_t$, calculated as $m_t = \beta_1 \cdot m_{t-1} + (1 - \beta_1) \cdot g_t$.
- $v_t$: EMA of the squared gradients, given by $v_t = \beta_2 \cdot v_{t-1} + (1 - \beta_2) \cdot g_t^2$.
- $\hat{m}_t, \hat{v}_t$: bias-corrected estimates of $m_t$ and $v_t$, respectively, where $\hat{m}_t = \frac{m_t}{1-\beta_1^t}$ and $\hat{v}_t = \frac{v_t}{1-\beta_2^t}$.
- $\alpha, \epsilon$: $\alpha$ is the learning rate, typically set to $10^{-3}$, and $\epsilon$ is a small constant to prevent division by zero, typically set to $10^{-8}$.
- $\beta_1, \beta_2$: smoothing parameters, commonly set as $\beta_1 = 0.9, \beta_2 = 0.999$.
- $\theta_t$: the parameter vector at step $t$.
- $\theta_0$: the initial parameter vector.

**Comparison with Adam** CAdam (Algorithm 1) and Adam both use the first and second moments of gradients to adapt learning rates. The main difference between CAdam and Adam is that CAdam introduces the alignment between the momentum and the gradient as a confidence metric to address two common problems in real-world online learning: distribution shifts and noise.

In Adam, the update direction is determined by $m_t$, the exponential moving average (EMA) of the gradient $g_t$, and $v_t$, the EMA of the squared gradients $g_t^2$. This method assumes a relatively stable data distribution, where $m_t$ serves as a good estimator of the optimal update direction. However, if the data distribution changes, $m_t$ may no longer point in the correct direction. Adam will continue to update using the outdated $m_t$ for several iterations until it eventually aligns with the new gradient

---

**Algorithm 1** Confidence Adaptive Moment Estimation (CAdam)

---

1: $m_0 \leftarrow 0, v_0 \leftarrow 0, \hat{v}_{\max,0} \leftarrow 0, t \leftarrow 0, \theta_t = \theta_0$
2: **while** $\theta_t$ not converged **do**
3:     $t \leftarrow t + 1$
4:     $g_t \leftarrow \nabla_\theta f_t(\theta_{t-1})$
5:     $m_t \leftarrow \beta_1 \cdot m_{t-1} + (1 - \beta_1) \cdot g_t$
6:     $v_t \leftarrow \beta_2 \cdot v_{t-1} + (1 - \beta_2) \cdot g_t^2$
7:     $\hat{m}_t \leftarrow m_t/(1 - \beta_1^t)$
8:     $\hat{v}_t \leftarrow v_t/(1 - \beta_2^t)$
9:     **if** AMSGrad **then**
10:         $\hat{v}_{\max,t} \leftarrow \max(\hat{v}_{\max,t-1}, \hat{v}_t)$
11:     **else**
12:         $\hat{v}_{\max,t} \leftarrow \hat{v}_t$
13:     **end if**
14:     $\hat{m}_t \leftarrow \max(0, m_t \cdot \text{sign}(g_t))$       ▷ Element-wise mask out elements where $m_t \cdot g_t \leq 0$
15:     $\theta_t \leftarrow \theta_{t-1} - \alpha \cdot \hat{m}_t/(\sqrt{\hat{v}_{\max,t}} + \epsilon)$
16: **end while**
17: **return** $\theta_t$

---

direction, leading to poor performance during this adaptation period. Additionally, when encountering noisy examples, Adam blindly updates using $m_t$, which can be problematic as it equivalently increases the learning rate, especially when the proportion of noisy data is high.

In contrast, CAdam dynamically checks the *alignment* between the current gradient $g_t$ and the momentum $m_t$ before proceeding with an update. If $g_t$ and $m_t$ point in the same direction, indicating that the momentum aligns with the current gradient, CAdam performs the update using $m_t/\sqrt{v_t}$. However, if $g_t$ and $m_t$ point in opposite directions, CAdam **pauses** the update for that parameter to observe subsequent gradients. This pause allows CAdam to distinguish between a potential distribution shift and noise.

If the reverse gradient signs persist in subsequent steps, it signals a distribution shift, and $m_t$ will gradually change direction to reflect the new data pattern, while CAdam doesn't update in these iterations, avoiding incorrect updates. Conversely, if the gradient signs realign in the following steps, it indicates that the previous opposite gradient was caused by noise. In this case, CAdam resumes normal updates, effectively filtering out noisy gradients without making unnecessary updates in the process.

In addition, CAdam also has an AMSGrad (Reddi et al., 2018) variant as described in 1 when AMSGrad option is enabled.

**Convergence Analysis**   Given a stream of functions $f_t : \mathbb{R}^d \to \mathbb{R}, t = 1, 2, \ldots, T$, an online learning algorithm chooses $\theta_t$ in each time step $t$ and aims to minimize the $T$-step regret w.r.t. the optimum, where the regret is defined as

$$R_T := \sum_{t=1}^T f_t(\theta_t) - \sum_{t=1}^T f_t(\theta^*), \quad \theta^* = \arg\min_\theta \sum_{t=1}^T f_t(\theta). \tag{1}$$

The online learning setting has been widely used to model real-world recommendation scenarios. We show that CAdam has the same $O(\sqrt{T})$ regret as Adam/AMSGrad under the same assumptions made in Reddi et al. (2018). The detailed proofs can be found in the appendix.

**Theorem 1** (Informal). *Under the assumptions introduced in Reddi et al. (2018), the CAdam algorithm (with AMSGrad correction) achieves a sublinear regret; that is,*

$$R_T = \mathcal{O}(\sqrt{T}). \tag{2}$$

**Remark:** We follow the regret analysis in Reddi et al. (2018) and adopt the same set of assumptions. In particular, Reddi et al. (2018) only considered convex functions and made bounded gradient assumption. Recently, there is a body of work that has provided refined convergence analysis under

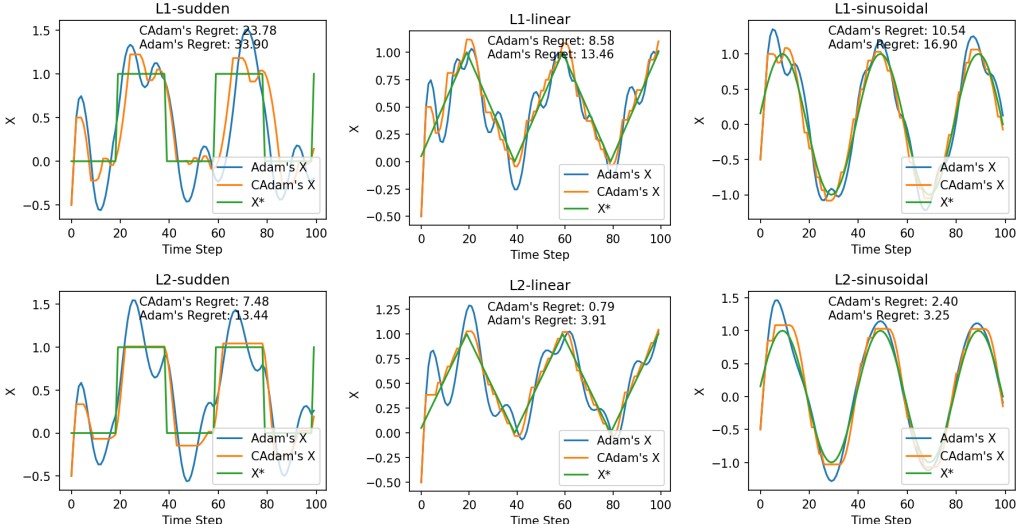

Figure 1: Trajectory of Adam (top row) and CAdam (bottom row) under different distribution changes: (Left) sudden change, (Middle) linear change, and (Right) sinusoidal change. The first row corresponds to the L1 loss landscapes, while the second row corresponds to the L2 loss landscapes. Adam's $X$ and CAdam's $X$ denote the locations of the optimization trajectories for Adam and CAdam, respectively, while $X^*$ represents the location of the optimal solution. CAdam shows superior adaptability to distribution shifts.

nonconvex setting and much weaker assumptions (see e.g., Alacaoglu et al. (2020); Défossez et al.; Zhang et al. (2022); Wang et al. (2024)). We leave the analysis of C-Adam under these more general settings as an interesting future direction.

## 4 EXPERIMENT

In this section, we systematically evaluate the performance of CAdam across various scenarios, starting with synthetic image data, followed by tests on a public advertisement dataset, and concluding with A/B tests in a real-world recommendation system. We first examine CAdam's behaviour under distribution shift, and noisy conditions using the CIFAR-10 dataset(Krizhevsky et al., 2009) with the VGG network(Simonyan & Zisserman, 2014). Next, we test CAdam against other popular optimizers on the Criteo dataset(Jean-Baptiste Tien, 2014), focusing on different models and scenarios. Finally, we conduct A/B tests with millions of users in a real-world recommendation system to validate CAdam's effectiveness in large-scale, production-level environments. The results demonstrate that CAdam consistently outperforms Adam and other optimizers across different tasks, distribution shifts, and noise conditions.

### 4.1 NUMERICAL EXPERIMENT

**Distribution Change** To illustrate the different behaviours of Adam and CAdam under distribution shifts, we designed three types of distribution changes for both L1 and L2 loss functions: (1) *Sudden* change, where the minimum shifts abruptly at regular intervals; (2) *Linear* change, where the minimum moves at a constant speed; and (3) *Sinusoidal* change, where the minimum oscillates following a sine function, resulting in variable speed over time.

The loss functions are defined as:

$$L(x,t) = \begin{cases} |x - x^*(t)|, & \text{L1 loss}, \\ (x - x^*(t))^2, & \text{L2 loss}, \end{cases}$$

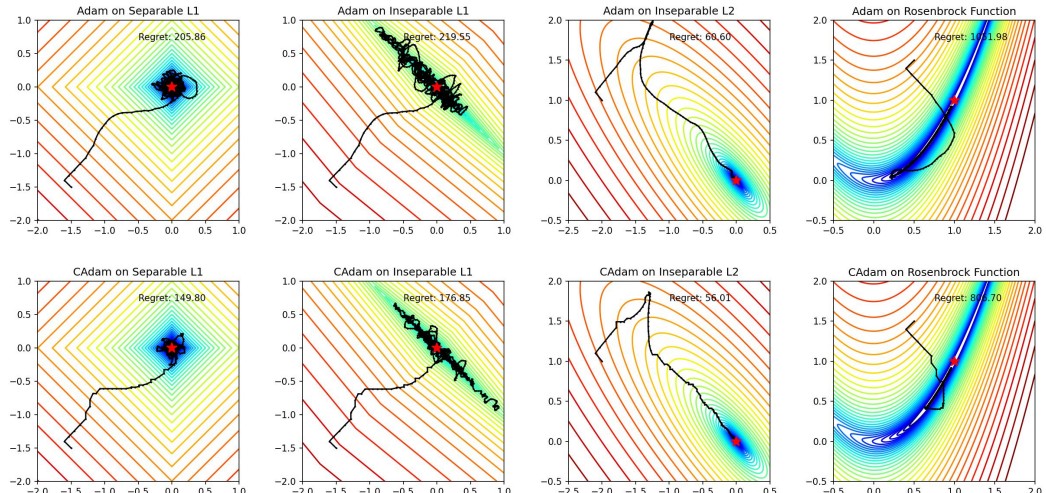

Figure 2: Trajectory of Adam (top row) and CAdam (bottom row) under noisy conditions on four different optimization landscapes: (Left to Right) separable L1 loss, inseparable L1 loss, inseparable L2 loss, and Rosenbrock function. Each column shows the optimization trajectory in the presence of noise, where each dimension's gradient is randomly flipped with a 50% probability. CAdam demonstrates superior robustness, maintaining more stable convergence paths than Adam across all tested functions.

where $x^*(t)$ represents the position of the minimum at time $t$ and is defined based on the type of distribution change:

$$x^*(t) = \begin{cases} \lfloor \frac{t}{T} \rfloor \mod 2, & \textit{sudden change}, \\ \frac{t}{T}, & \textit{linear change}, \\ \sin\left(\frac{2\pi t}{T}\right), & \textit{sinusoidal change}. \end{cases}$$

The results of these experiments are presented in Figure 1. Across different loss functions and distribution changes, CAdam closely follows the trajectory of the minimum point, being less affected by incorrect momentum, exhibiting lower regret and demonstrating its superior ability to adapt to shifting distributions.

**Noisy Samples**  To compare Adam and CAdam in noisy environments, we conducted experiments on four different optimization 2-d landscapes: (1) separable L1 loss, (2) inseparable L1 loss, (3) inseparable L2 loss, and (4) Rosenbrock function. These landscapes are defined as follows:

1. Separable L1 Loss: $f_1(x, y) = |x| + |y|$.

2. Inseparable L1 Loss: $f_2(x, y) = |x + y| + \frac{|x-y|}{10}$.

3. Inseparable L2 Loss: $f_3(x, y) = (x + y)^2 + \frac{(x-y)^2}{10}$.

4. Rosenbrock Function: $f_4(x, y) = (a - x)^2 + b(y - x^2)^2$, where $a = 1$ and $b = 100$.

To simulate noise in the gradients, we applied a random mask to each dimension of the gradient with a 50% probability using the same random seed across different optimizers. Specifically, the gradient components were multiplied by a uniformly distributed random value from the range $[-1, 1]$ to introduce noise:

$$\nabla_{\text{noisy}}(x, y) = \begin{cases} \nabla f(x, y) \cdot U(-1, 1), & \text{with probability } p = 0.5, \\ \nabla f(x, y), & \text{otherwise}, \end{cases}$$

The results of these experiments are shown in Figure 2. For comparison, the results without noise are provided in Figure 5 in the appendix. The trajectory of CAdam exhibits fewer random perturbations and lower regret, indicating its ability to resist noise interference.

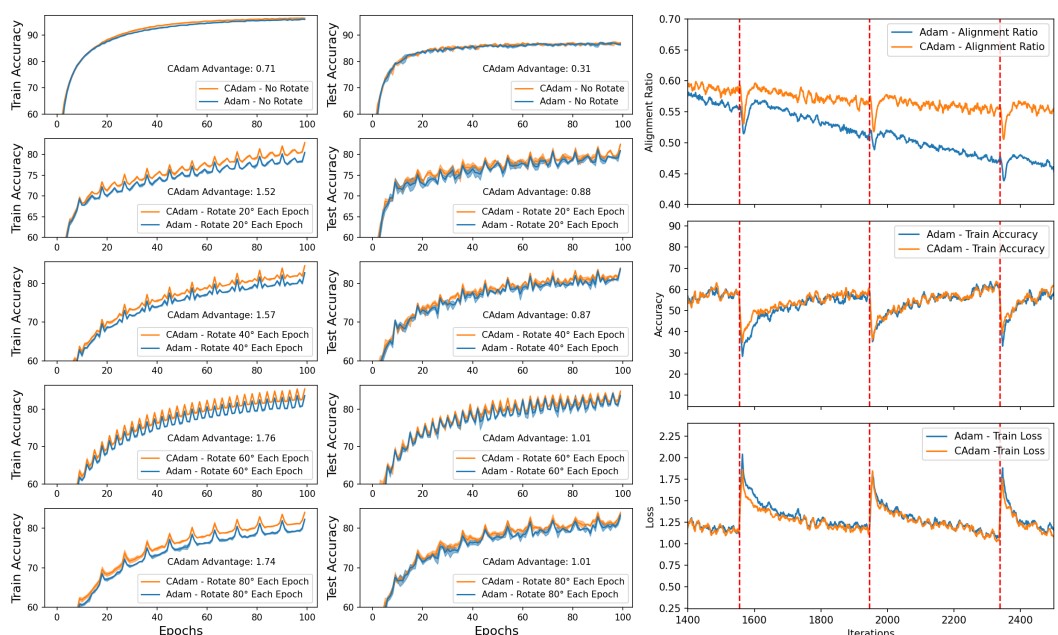

Figure 3: **(Left)** Performance of CAdam and Adam under different rotation speeds corresponding to sudden distribution shift. CAdam demonstrates superior performance, with a more pronounced advantage over Adam in the presence of rotation. **(Right)** A detailed view at a 60-degree rotation between steps 1400 to 2300, showing the Alignment Ratio, Accuracy, and Loss. The red dashed lines indicate the rotation points, where the alignment ratio decreases, resulting in fewer parameter updates. This is followed by a gradual recovery in both the alignment ratio and accuracy, and a decline in loss. CAdam's accuracy drop is slower, and its recovery is faster than Adam's, illustrating its enhanced ability to adapt to distribution shifts.

## 4.2 CNN ON IMAGE CLASSIFICATION

We perform experiments using the VGG network on the CIFAR-10 dataset to evaluate the effectiveness of CAdam in handling distribution shifts and noise. We synthesize three experimental conditions: (1) sudden distribution changes, (2) continuous distribution shifts, and (3) added noise to the samples. The hyperparameters for these experiments are provided in Section B.2.

**Sudden Distribution Shift**   To simulate sudden changes in data distribution, we rotate the images by a specific angle at the start of each epoch, relative to the previous epoch, as illustrated in Figure 3. CAdam consistently outperforms Adam across varying rotation speeds, with a more significant performance gap compared to the non-rotated condition.

We define the *alignment ratio* as:

$$\text{Alignment Ratio} = \frac{\text{Number of parameters where } m_t \cdot g_t > 0}{\text{Total number of parameters}}$$

A closer inspection in Figure 3 reveals that, during the rotation (indicated by the red dashed line), the alignment ratio decreases, resulting in fewer parameters being updated, followed by a gradual recovery. Correspondingly, the accuracy declines and subsequently improves, while the loss increases before decreasing. Notably, during these shifts, CAdam's accuracy drops more slowly and recovers faster than Adam's, indicating its superior adaptability to new data distributions.

**Continuous Distribution Shifts**   In contrast to sudden distribution changes, we also tested the scenario where the data distribution changes continuously. Specifically, we simulated this by rotating the data distribution at each iteration by an angle. The results, shown in Figure 4, indicate that as the rotation speed increases, the advantage of CAdam over Adam becomes more pronounced.

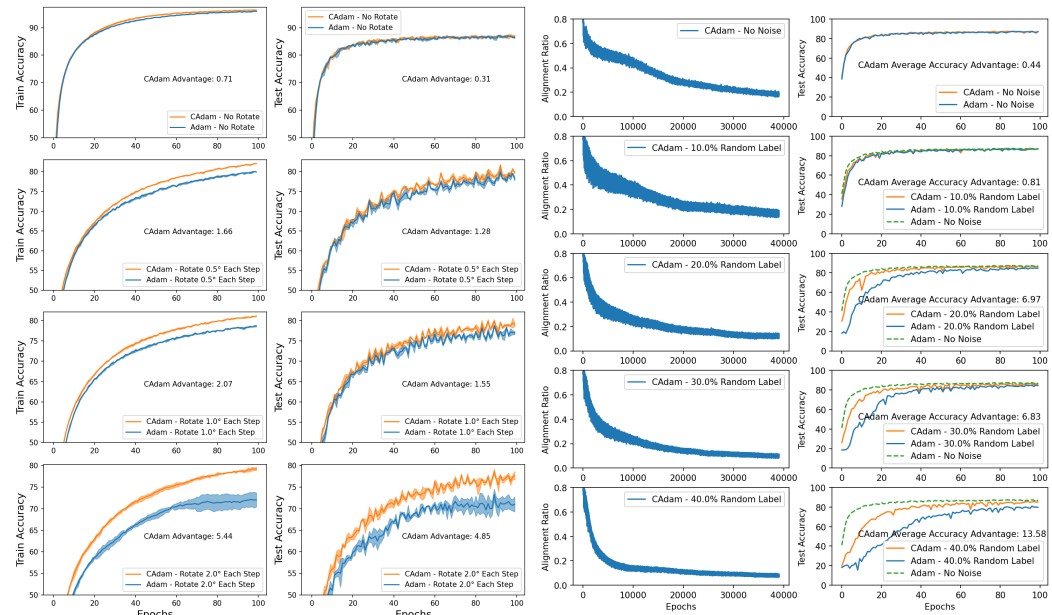

Figure 4: **(Left)** Performance of CAdam and Adam under continuous distribution shifts with different rotation speeds. CAdam demonstrates superior performance, with its advantage becoming more pronounced as the rotation speed increases. **(Right)** The effect of adding noise to the samples. CAdam exhibits a slower accuracy drop compared to Adam, showcasing its enhanced robustness to noisy data.

**Noisy Samples** To evaluate the optimizer's robustness to noise, we introduced noise into the dataset by randomly selecting a certain number of batches in each epoch (resampling for each epoch) and replacing the labels of these batches with random values. The results are presented in Figure 4. We observed that as the proportion of noisy labels increases, the consistency of CAdam decreases, causing it to update fewer parameters in each iteration. Despite this, both CAdam and Adam experience a performance decline in test set accuracy as noise increases. Nevertheless, CAdam consistently outperforms Adam, maintaining accuracy even with 40% noise, comparable to Adam's performance in a noise-free setting by the end of training.

### 4.3 PUBLIC ADVERTISEMENT DATASET

**Experiment Setting** To evaluate the effectiveness of the proposed CAdam optimizer, we conducted experiments using various models on the Criteo-x4-001 dataset(Jean-Baptiste Tien, 2014). This dataset contains feature values and click feedback for millions of display ads and is commonly used to benchmark algorithms for click-through rate (CTR) prediction(Zhu et al., 2021). To simulate a real-world online learning scenario, we trained the models on data up to each timestamp in a single epoch(Fukushima et al., 2020). This setup replicates the environment where new data arrives continuously, requiring the model to adapt quickly.

Furthermore, for sparse parameters (e.g., embeddings), we update the optimizer's state only when there is a non-zero gradient for this parameter in the current batch using SparseAdam implementation in Pytorch(Paszke et al., 2019). This approach ensures that the optimizer's state reflects the parameters influenced by recent data changes. The hyperparameters are provided in Appendix B.3.

We benchmarked CAdam and other popular optimizers, including SGD, SGDM(Qian, 1999), AdaGrad(Duchi et al., 2011), AdaDelta(Zeiler, 2012), RMSProp, Adam(Kingma & Ba, 2015), AMSGrad(Reddi et al., 2018), and AdaBelief(Zhuang et al., 2020), on various models such as DeepFM(77M)(Guo et al., 2017), WideDeep(77M)(Cheng et al., 2016), DNN(74M)(Covington et al., 2016), PNN(79M)(Qu et al., 2016), and DCN(74M)(Wang et al., 2017). The performance of these optimizers was evaluated using the Area Under the Curve (AUC) metric.

Table 1: AUC performance of different optimizers on the Criteo dataset across various models. Results are averaged over three seeds with mean and standard deviation ($\pm$) reported. CAmsGrad denotes the AMSGrad variant of CAdam, which achieves the highest average performance.

| | DeepFM | WideDeep | DNN | PNN | DCN | Avg |
|---|---|---|---|---|---|---|
| SGD | $71.90_{\pm.006}$ | $71.88_{\pm.013}$ | $68.12_{\pm.043}$ | $67.61_{\pm.318}$ | $69.55_{\pm.026}$ | 69.81 |
| SGDM | $76.59_{\pm.044}$ | $76.32_{\pm.021}$ | $78.80_{\pm.014}$ | $76.17_{\pm.050}$ | $77.90_{\pm.018}$ | 77.16 |
| AdaGrad | $71.77_{\pm.032}$ | $71.50_{\pm.011}$ | $68.65_{\pm.022}$ | $67.49_{\pm.027}$ | $69.55_{\pm.020}$ | 69.79 |
| AdaDelta | $71.91_{\pm.071}$ | $71.64_{\pm.005}$ | $69.76_{\pm.004}$ | $67.59_{\pm.025}$ | $69.76_{\pm.024}$ | 70.13 |
| RMSProp | $71.82_{\pm.010}$ | $71.54_{\pm.021}$ | $68.72_{\pm.005}$ | $67.51_{\pm.004}$ | $69.60_{\pm.007}$ | 69.84 |
| Adam | $80.87_{\pm.011}$ | $80.90_{\pm.004}$ | $80.89_{\pm.003}$ | $80.90_{\pm.006}$ | $81.05_{\pm.005}$ | 80.92 |
| AdaBelief | $80.84_{\pm.008}$ | $80.90_{\pm.002}$ | $80.88_{\pm.011}$ | $80.89_{\pm.002}$ | $81.02_{\pm.044}$ | 80.91 |
| AdamW | $80.87_{\pm.008}$ | $80.90_{\pm.010}$ | $80.88_{\pm.010}$ | $80.90_{\pm.002}$ | $81.00_{\pm.047}$ | 80.91 |
| AmsGrad | $80.88_{\pm.004}$ | $80.92_{\pm.008}$ | $80.91_{\pm.001}$ | $80.92_{\pm.009}$ | $81.08_{\pm.009}$ | 80.94 |
| CAdam | $80.88_{\pm.008}$ | $80.93_{\pm.004}$ | $80.90_{\pm.002}$ | $80.93_{\pm.006}$ | $81.06_{\pm.009}$ | 80.94 |
| CAmsGrad | $\mathbf{80.90_{\pm.006}}$ | $\mathbf{80.93_{\pm.007}}$ | $\mathbf{80.92_{\pm.005}}$ | $\mathbf{80.94_{\pm.009}}$ | $\mathbf{81.09_{\pm.010}}$ | **80.96** |

**Main Results**    The results in Table 1 show that CAdam and its AMSGrad variants outperform other optimizers across different models. While the AMSGrad variants perform better on certain datasets, they do not consistently outperform standard CAdam. Both versions of CAdam generally achieve higher AUC scores than other optimizers, demonstrating their effectiveness in the online learning setting.

**Robustness under Noise**    To simulate a noisier environment, we introduced noise into the Criteo x4-001 dataset by flipping 1% of the negative training samples to positive. All other settings remained unchanged. The results in Table 2 show that CAdam consistently outperforms Adam in terms of both AUC and the extent of performance drop. This demonstrates CAdam's robustness in handling noisy data.

Table 2: Results of Adam and CAdam on the Noisy Criteo dataset, averaged over three seeds. "Drop" indicates the decrease in performance compared to training on the original Criteo dataset. CAdam shows a smaller performance drop, highlighting its robustness to noise.

| | DeepFM | WideDeep | DNN | PNN | DCN |
|---|---|---|---|---|---|
| Adam | $80.51_{\pm.008}$ | $80.47_{\pm.006}$ | $80.48_{\pm.014}$ | $80.66_{\pm.006}$ | $80.51_{\pm.010}$ |
| CAdam | $\mathbf{80.81_{\pm.007}}$ | $\mathbf{80.79_{\pm.006}}$ | $\mathbf{80.78_{\pm.005}}$ | $\mathbf{80.96_{\pm.026}}$ | $\mathbf{80.77_{\pm.007}}$ |
| Adam Drop | $-0.36_{\pm.014}$ | $-0.43_{\pm.007}$ | $-0.41_{\pm.016}$ | $-0.23_{\pm.012}$ | $-0.54_{\pm.013}$ |
| CAdam Drop | $\mathbf{-0.08_{\pm.014}}$ | $\mathbf{-0.14_{\pm.009}}$ | $\mathbf{-0.12_{\pm.004}}$ | $\mathbf{+0.04_{\pm.031}}$ | $\mathbf{-0.28_{\pm.015}}$ |

## 4.4 Experiment on Real-World Recommendation System

In real-world recommendation scenarios, the differences from the Criteo dataset experiments are quite significant. First, both data volume and model sizes are much larger, with models used in the following experiments ranging from 8.3 billion to 330 billion parameters—100 to 10,000 times larger. Second, as these are online experiments, unlike offline experiments with a fixed dataset, the model's output directly influences user behaviour. To test the effectiveness of CAdam in this setting, we conducted A/B tests on internal models serving millions of users across seven different scenarios (2 pre-ranking, 4 recall, and 1 ranking).

During these online experiments, we used a batch size of $B = 4096$ The evaluation metric was the Generalized Area Under the Curve (GAUC). Due to limited resources, we compared only Adam and CAdam, running the experiments for 48 hours.

The results, shown in Table 3, indicate that CAdam consistently outperformed Adam across all test scenarios, demonstrating its superiority in real-world applications.

Table 3: GAUC results for Adam and CAdam across seven internal experiment settings. "Pr" denotes pre-ranking, "Rec" represents recall, and "Rk" indicates ranking. CAdam consistently outperforms Adam, highlighting its effectiveness in real-world recommendation scenarios.

| Metric | Pr 1 | Pr 2 | Rec 1 | Rec 2 | Rec 3 | Rec 4 | Rk 1 | Average |
|--------|-------|-------|-------|-------|-------|-------|-------|---------|
| **Adam** | 87.41% | 82.89% | 90.18% | 82.41% | 84.57% | 85.39% | 88.52% | 85.34% |
| **CAdam** | 87.61% | 83.28% | 90.43% | 82.61% | 85.06% | 85.49% | 88.74% | 85.64% |
| **Impr.** | 0.20% | 0.39% | 0.25% | 0.20% | 0.49% | 0.10% | 0.22% | 0.30% |

## 5 CONCLUSION

In this paper, we addressed the inherent limitations of the Adam optimizer in online learning environments, particularly its sluggish adaptation to distributional shifts and heightened sensitivity to noisy data. To overcome these challenges, we introduced CAdam (Confidence Adaptive Moment Estimation), a novel optimization strategy that enhances Adam by incorporating a confidence-based mechanism. This mechanism evaluates the alignment between momentum and gradients for each parameter dimension, ensuring that updates are performed judiciously. When momentum and gradients are aligned, CAdam updates the parameters following Adam's original formulation; otherwise, it temporarily withholds updates to discern between true distribution shifts and transient noise.

Our extensive experiments across synthetic benchmarks, public advertisement datasets, and large-scale real-world recommendation systems consistently demonstrated that CAdam outperforms Adam and other well-established optimizers in both adaptability and robustness. Specifically, CAdam showed superior performance in scenarios with sudden and continuous distribution shifts, as well as in environments with significant noise, achieving higher accuracy and lower regret. Moreover, in live A/B testing within a production recommendation system, CAdam led to substantial improvements in model performance and gross merchandise volume (GMV), underscoring its practical effectiveness.

Future work may explore further refinements of the confidence assessment mechanism, its integration with other optimization frameworks, and its application to a broader range of machine learning models and real-time systems. Ultimately, CAdam represents a promising advancement in the development of more resilient and adaptive optimization algorithms for dynamic learning environments.

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

## A    PROOFS OF THEOREM 1

Given a stream of objectives $f_t : \mathbb{R}^d \to \mathbb{R}, t = 1, 2, \ldots, T$, online learning aims to minimize the regret w.r.t. the optimum; that is,

$$R_T := \sum_{t=1}^{T} f_t(x_t) - \sum_{t=1}^{T} f_t(x^*), \quad x^* = \underset{x}{\arg\min} \sum_{t=1}^{T} f_t(x). \tag{3}$$

Recall that each update in CAdam can be characterized as follows[1]:

$$m_t = \beta_{1,t} m_{t-1} + (1 - \beta_{1,t}) g_t, \tag{4}$$

$$v_t = \beta_2 v_{t-1} + (1 - \beta_2) g_t^2, \tag{5}$$

$$m_{t,\Xi_t} = \begin{cases} m_{t,i}, & i \in \Xi_t \\ 0, & \text{else} \end{cases}, \tag{6}$$

$$\hat{v}_t = \max(\hat{v}_{t-1}, v_t), \tag{7}$$

$$x_{t+1} = x_t - \alpha_t m_{t,\Xi_t} / \hat{v}_t. \tag{8}$$

where $\Xi_t := \{i \in [d] : m_{t,i} \cdot g_{t,i} \geq 0\}$ indicates the set of active entries at step $t$. For notation clarity, let $x_{t,\Xi}$ be the vector of which the entries not belonging to $\Xi$ are masked. Following the AMSGrad (Reddi et al., 2018), we are to prove that the sequence of points obtained by CAdam satisfies $R_T/T \to 0$ as $T$ increases.

We first introduce three standard assumptions:

**Assumption 1.** *Let $f_t : \mathbb{R}^d \to \mathbb{R}, t = 1, 2, \ldots, T$ be a sequence of convex and differentiable functions with $\|\nabla f_t(x)\|_\infty \leq G_\infty$ for all $t \in [T]$.*

**Assumption 2.** *Let $\{m_t\}, \{v_t\}$ be the sequences used in CAdam, $\alpha_t = \alpha/\sqrt{t}, \beta_{1,t} = \beta_1 \lambda^{t-1} < 1, \gamma = \beta_1/\sqrt{\beta_2} < 1$ for all $t \in [T]$.*

**Assumption 3.** *The points involved are within a bounded diameter $D_\infty$; that is, for the optimal point $x^*$ and any points $x_t$ generated by CAdam, it holds $\|x_t - x^*\|_\infty \leq D_\infty/2$.*

We present several essential lemmas in the following. Given that some of these lemmas have been partially established in prior works (Kingma & Ba, 2015; Reddi et al., 2018), we include them here for the sake of completeness.

**Lemma 1.** *For a convex and differentiable function $f : \mathbb{R}^d \to \mathbb{R}$, we have*

$$f(x) - f(y) \leq \langle \nabla f(x), x - y \rangle. \tag{9}$$

**Lemma 2.** *Under Assumption 1 and 2, we have*

$$\begin{aligned}
\left\langle g_{t,\Xi_t}, x_{t,\Xi_t} - x^*_{\Xi_t} \right\rangle \leq \quad & \frac{1}{2\alpha_t(1-\beta_{1,t})} \left( \|V_t^{1/4}(x_{t,\Xi_t} - x^*_{\Xi_t})\|^2 - \|V_t^{1/4}(x_{t+1,\Xi_t} - x^*_{\Xi_t})\|^2 \right) \\
& + \frac{\beta_{1,t}}{2\alpha_t(1-\beta_{1,t})} \|V_t^{1/4}(x_t - x^*)\|^2 \\
& + \frac{\alpha_t}{2(1-\beta_{1,t})} \|V_t^{-1/4} m_t\|^2 + \frac{\alpha_t \beta_{1,t}}{2(1-\beta_{1,t})} \|V_t^{-1/4} m_{t-1}\|^2,
\end{aligned} \tag{10}$$

*where $V_t := \text{diag}(\hat{v}_t)$.*

*Proof.* CAdam updates the parameters as follows

$$x_{t+1,\Xi_t} = x_{t,\Xi_t} - \alpha_t m_{t,\Xi_t} / \sqrt{\hat{v}_t} = x_{t,\Xi_t} - \alpha_t V_t^{-1/2} \Big( \beta_{1,t} m_{t-1,\Xi_t} + (1-\beta_{1,t}) g_{t,\Xi_t} \Big).$$

Subtracting $x^*$ from both sides yields

$$\|V_t^{1/4}(x_{t+1,\Xi_t} - x^*_{\Xi_t})\|_2^2$$

$$= \|V_t^{1/4}(x_{t,\Xi_t} - x^*_{\Xi_t}) - \alpha_t V_t^{-1/4} m_{t,\Xi_t}\|_2^2$$

$$= \|V_t^{1/4}(x_{t,\Xi_t} - x^*_{\Xi_t})\|_2^2 - 2\langle \alpha_t V_t^{-1/4} m_{t,\Xi_t}, V_t^{1/4}(x_{t,\Xi_t} - x^*_{\Xi_t}) \rangle + \|\alpha_t V_t^{-1/4} m_{t,\Xi_t}\|_2^2$$

$$= \|V_t^{1/4}(x_{t,\Xi_t} - x^*_{\Xi_t})\|_2^2 - 2\alpha_t \langle \beta_{1,t} m_{t-1,\Xi_t} + (1-\beta_{1,t}) g_{t,\Xi_t}, x_{t,\Xi_t} - x^*_{\Xi_t} \rangle + \|\alpha_t V_t^{-1/4} m_{t,\Xi_t}\|_2^2.$$

---

[1] Note that we omit the bias corrections for clarity purpose. It is not difficult to modify the proofs to obtain a more general one.

Rearranging the equation gives

$$
\left\langle g_{t,\Xi_t}, x_{t,\Xi_t} - x^*_{\Xi_t} \right\rangle = \frac{1}{2\alpha_t(1-\beta_{1,t})}\left( \|V_t^{1/4}(x_{t,\Xi_t} - x^*_{\Xi_t})\|_2^2 - \|V_t^{1/4}(x_{t+1,\Xi_t} - x^*_{\Xi_t})\|_2^2 \right)
$$
$$
- \frac{\beta_{1,t}}{1-\beta_{1,t}}\left\langle m_{t-1,\Xi_t}, x_{t,\Xi_t} - x^*_{\Xi_t} \right\rangle + \frac{\alpha_t}{2(1-\beta_{1,t})}\|V_t^{-1/4}m_{t,\Xi_t}\|_2^2.
$$

The results follow from the Cauchy-Schwarz inequality and Young's inequality:

$$
-\frac{\beta_{1,t}}{1-\beta_{1,t}}\left\langle m_{t-1,\Xi_t}, x_{t,\Xi_t} - x^*_{\Xi_t} \right\rangle = \frac{\beta_{1,t}}{1-\beta_{1,t}}\left\langle m_{t-1,\Xi_t}, x^*_{\Xi_t} - x_{t,\Xi_t} \right\rangle
$$
$$
= \frac{\beta_{1,t}}{1-\beta_{1,t}}\left\langle \sqrt{\alpha_t}V_t^{-1/4}m_{t-1,\Xi_t}, \frac{1}{\sqrt{\alpha_t}}V_t^{1/4}(x^*_{\Xi_t} - x_{t,\Xi_t}) \right\rangle
$$
$$
\leq \frac{\beta_{1,t}}{1-\beta_{1,t}}\left( \sqrt{\alpha_t}\|V_t^{-1/4}m_{t-1,\Xi_t}\| \cdot \frac{1}{\sqrt{\alpha_t}}\|V_t^{1/4}(x^*_{\Xi_t} - x_{t,\Xi_t})\| \right)
$$
$$
\leq \frac{\beta_{1,t}}{1-\beta_{1,t}}\left( \frac{\alpha_t}{2}\|V_t^{-1/4}m_{t-1,\Xi_t}\|^2 + \frac{1}{2\alpha_t}\|V_t^{1/4}(x_{t,\Xi_t} - x^*_{\Xi_t})\|^2 \right)
$$
$$
\leq \frac{\beta_{1,t}}{1-\beta_{1,t}}\left( \frac{\alpha_t}{2}\|V_t^{-1/4}m_{t-1}\|^2 + \frac{1}{2\alpha_t}\|V_t^{1/4}(x_t - x^*)\|^2 \right),
$$

and the fact that $\|V_t^{-1/4}m_{t,\Xi_t}\|_2^2 \leq \|V_t^{-1/4}m_t\|_2^2$.

$\square$

**Lemma 3.** *Under Assumption 1, 2, and 3, we have*

$$
\left\langle g_t, x_t - x^* \right\rangle \leq \left\langle g_{t,\Xi}, x_{t,\Xi} - x^*_{\Xi} \right\rangle + \frac{d\beta_1\lambda^{t-1}D_\infty G_\infty}{1-\beta_1}. \tag{11}
$$

*Proof.* If the $i$-th entry is not updated at step $t$, i.e., $i \in [d] \setminus \Xi_t$, it can be derived that

$$
\left( \beta_{1,t}m_{t-1,i} + (1-\beta_{1,t})g_{t,i} \right) \cdot g_{t,i} \leq 0
$$
$$
\Rightarrow \left( \beta_{1,t}m_{t-1,i} + (1-\beta_{1,t})g_{t,i} \right) \cdot \mathrm{sgn}(g_{t,i}) \leq 0
$$
$$
\Rightarrow -\beta_{1,t}|m_{t-1,i}| + (1-\beta_{1,t})|g_{t,i}| \leq 0
$$
$$
\Rightarrow |g_{t,i}| \leq \frac{\beta_{1,t}}{1-\beta_{1,t}}|m_{t-1,i}|
$$
$$
\Rightarrow |g_{t,i}| \leq \frac{\beta_{1,t}}{1-\beta_{1,t}}G_\infty \qquad\qquad \leftarrow \text{Assumption 1}
$$
$$
\Rightarrow |g_{t,i}| \leq \frac{\beta_1\lambda^{t-1}}{1-\beta_1}G_\infty, \quad i \in [d]\setminus\Xi_t. \qquad \leftarrow \text{Assumption 2}
$$

With Assumption 3, it immediately yields the desired inequality that

$$
\left\langle g_t, x_t - x^* \right\rangle = \left\langle g_{t,\Xi}, x_{t,\Xi} - x^*_{\Xi} \right\rangle + \left\langle g_{t,[d]\setminus\Xi}, x_{t,[d]\setminus\Xi} - x^*_{[d]\setminus\Xi} \right\rangle
$$
$$
\leq \left\langle g_{t,\Xi}, x_{t,\Xi} - x^*_{\Xi} \right\rangle + \sum_{i=1}^{d}\frac{\beta_1\lambda^{t-1}D_\infty G_\infty}{1-\beta_1}.
$$

$\square$

**Lemma 4.** *Given Assumption 1, 2, and 3, we have*

$$
\sum_{t\in[T]}\frac{\beta_{1,t}}{2\alpha_t(1-\beta_{1,t})}\|V_t^{1/4}(x_t - x^*)\|^2 \leq \frac{dD_\infty^2 G_\infty}{2\alpha(1-\beta_1)(1-\lambda)^2}. \tag{12}
$$

*Proof.*

$$\sum_{t\in[T]}\frac{\beta_{1,t}}{2\alpha_t(1-\beta_{1,t})}\|V_t^{1/4}(x_t-x^*)\|^2$$

$$\leq\frac{1}{2\alpha(1-\beta_1)}\sum_{t\in[T]}\sqrt{t}\lambda^{t-1}\|V_t^{1/4}(x_t-x^*)\|^2$$

$$\leq\frac{G_\infty}{2\alpha(1-\beta_1)}\sum_{t\in[T]}\sqrt{t}\lambda^{t-1}\|x_t-x^*\|^2 \qquad\qquad \leftarrow \text{Assumption 1}$$

$$\leq\frac{dD_\infty^2 G_\infty}{2\alpha(1-\beta_1)}\sum_{t\in[T]}\sqrt{t}\lambda^{t-1} \qquad\qquad \leftarrow \text{Assumption 3}$$

$$\leq\frac{dD_\infty^2 G_\infty}{2\alpha(1-\beta_1)}\sum_{t\in[T]}\lambda^{t-1}t$$

$$\leq\frac{dD_\infty^2 G_\infty}{2\alpha(1-\beta_1)}\frac{1}{(1-\lambda)^2}.$$

$\square$

**Lemma 5** (Reddi et al. (2018) Lemma2). *Under Assumption 2, we have*

$$\sum_{t\in[T]}\alpha_t\|V_t^{-1/4}m_t\|^2 \leq \frac{\alpha dG_\infty}{(1-\gamma)(1-\beta_1)\sqrt{1-\beta_2}}\sqrt{T}, \tag{13}$$

*where* $\gamma := \beta_1/\sqrt{\beta_2}$.

We are ready to prove the final results now. Concretely, Theorem 1 is a straightfoward corollary of the following conclusion.

**Theorem 2.** *Under the Assumption 1, 2, and 3, the regret is converged with*

$$R_T \leq \frac{dD_\infty^2 G_\infty\sqrt{T}}{2\alpha(1-\beta_1)} + \frac{d(2\alpha+D_\infty)D_\infty G_\infty}{2\alpha(1-\beta_1)(1-\lambda)^2} + \frac{\alpha dG_\infty\sqrt{T}}{(1-\gamma)(1-\beta_1)^2\sqrt{1-\beta_2}}. \tag{14}$$

*Proof.* Based on Lemma 1, Lemma 2, and Lemma 3, the regret can be firstly bounded by

$$R_T = \sum_{t\in[T]}(f_t(x_t)-f_t(x^*)) \leq \sum_{t\in[T]}\langle g_t, x_t-x^*\rangle$$

$$\leq \sum_{t\in[T]}\langle g_{t,\Xi_t}, x_{t,\Xi_t}-x^*_{\Xi_t}\rangle + \sum_{t\in[T]}\frac{d\beta_1\lambda^{t-1}D_\infty G_\infty}{1-\beta_1}$$

$$\leq \underbrace{\sum_{t\in[T]}\frac{1}{2\alpha_t(1-\beta_{1,t})}\left(\|V_t^{1/4}(x_{t,\Xi_t}-x^*_{\Xi_t})\|^2-\|V_t^{1/4}(x_{t+1,\Xi_t}-x^*_{\Xi_t})\|^2\right)}_{①}$$

$$+ \underbrace{\sum_{t\in[T]}\frac{\beta_{1,t}}{2\alpha_t(1-\beta_{1,t})}\|V_t^{1/4}(x_t-x^*)\|^2}_{②} + \underbrace{\sum_{t\in[T]}\frac{\alpha_t}{2(1-\beta_{1,t})}\|V_t^{-1/4}m_t\|^2}_{③}$$

$$+ \underbrace{\sum_{t\in[T]}\frac{\alpha_t\beta_{1,t}}{2(1-\beta_{1,t})}\|V_t^{-1/4}m_{t-1}\|^2}_{④} + \underbrace{\sum_{t\in[T]}\frac{d\beta_1\lambda^{t-1}D_\infty G_\infty}{1-\beta_1}}_{⑤}.$$

Let us address each term in turn. For the first term, we are to separately bound each entry and the results follows from the summation. For the $i$-th entry, let $\mathcal{T}_+^i = [t : i \in \bar{\Xi}_t]$ be a sequence collecting

all steps that $x_i$ is succesfully updated, and $\tilde{t}_k \in \mathcal{T}_+^i$ be the $k$-th element of $\mathcal{T}_+^i$. For simplicity, we will omit the superscript without ambiguity.

$$\textcircled{1}_i = \sum_{t=\tilde{t}_1}^{\tilde{t}_{|\mathcal{T}_+|}} \frac{1}{2\alpha_t(1-\beta_{1,t})} \left( (\hat{v}_{t,i}^{1/4}(x_{t,i}-x_i^*))^2 - (\hat{v}_{t,i}^{1/4}(x_{t+1,i}-x_i^*))^2 \right)$$

$$\leq \frac{\hat{v}_{\tilde{t}_1,i}^{1/2}(x_{\tilde{t}_1,i}-x_i^*)^2}{2\alpha_{\tilde{t}_1}(1-\beta_1)} + \frac{1}{2}\sum_{t=\tilde{t}_2}^{\tilde{t}_{|\mathcal{T}_+|}} \left[ \frac{\hat{v}_{t,i}^{1/2}(x_{t,i}-x_i^*)^2}{\alpha_t(1-\beta_{1,t})} - \frac{\hat{v}_{t-1,i}^{1/2}(x_{t,i}-x_i^*)^2}{\alpha_{t-1}(1-\beta_{1,t-1})} \right]$$

$$= \frac{\hat{v}_{\tilde{t}_1,i}^{1/2}(x_{\tilde{t}_1,i}-x_i^*)^2}{2\alpha_{\tilde{t}_1}(1-\beta_1)} + \frac{1}{2}\sum_{t=\tilde{t}_2}^{\tilde{t}_{|\mathcal{T}_+|}} \left[ \frac{\hat{v}_{t,i}^{1/2}(x_{t,i}-x_i^*)^2}{\alpha_t(1-\beta_{1,t-1})} \underbrace{- \frac{\hat{v}_{t,i}^{1/2}(x_{t,i}-x_i^*)^2}{\alpha_t(1-\beta_{1,t-1})} + \frac{\hat{v}_{t,i}^{1/2}(x_{t,i}-x_i^*)^2}{\alpha_t(1-\beta_{1,t})}}_{\leq 0} \right.$$

$$\left. - \frac{\hat{v}_{t-1,i}^{1/2}(x_{t,i}-x_i^*)^2}{\alpha_{t-1}(1-\beta_{1,t-1})} \right]$$

$$\leq \frac{\hat{v}_{\tilde{t}_1,i}^{1/2}(x_{\tilde{t}_1,i}-x_i^*)^2}{2\alpha_{\tilde{t}_1}(1-\beta_1)} + \frac{1}{2}\sum_{t=\tilde{t}_2}^{\tilde{t}_{|\mathcal{T}_+|}} \underbrace{\frac{1}{1-\beta_{1,t-1}}}_{\leq 1/(1-\beta_1)} \underbrace{\left[ \frac{\hat{v}_{t,i}^{1/2}(x_{t,i}-x_i^*)^2}{\alpha_t} - \frac{\hat{v}_{t-1,i}^{1/2}(x_{t,i}-x_i^*)^2}{\alpha_{t-1}} \right]}_{\geq 0 \text{ by } \hat{v}_{t,i} \geq \hat{v}_{t-1,i}}$$

$$\leq \frac{\hat{v}_{\tilde{t}_1,i}^{1/2}(x_{\tilde{t}_1,i}-x_i^*)^2}{2\alpha_{\tilde{t}_1}(1-\beta_1)} + \frac{D_\infty^2}{2(1-\beta_1)}\sum_{t=\tilde{t}_2}^{\tilde{t}_{|\mathcal{T}_+|}} \left[ \frac{\hat{v}_{t,i}^{1/2}}{\alpha_t} - \frac{\hat{v}_{t-1,i}^{1/2}}{\alpha_{t-1}} \right] \qquad \leftarrow \text{Assumption 3}$$

$$= \frac{\hat{v}_{\tilde{t}_1,i}^{1/2}(x_{\tilde{t}_1,i}-x_i^*)^2}{2\alpha_{\tilde{t}_1}(1-\beta_1)} + \frac{D_\infty^2}{2(1-\beta_1)} \left[ \frac{\hat{v}_{\tilde{t}_{|\mathcal{T}_+|},i}^{1/2}}{\alpha_{\tilde{t}_{|\mathcal{T}_+|}}} - \frac{\hat{v}_{\tilde{t}_1,i}^{1/2}}{\alpha_{\tilde{t}_1}} \right]$$

$$\leq \frac{D_\infty^2}{2(1-\beta_1)} \frac{\hat{v}_{\tilde{t}_{|\mathcal{T}_+|},i}^{1/2}}{\alpha_{\tilde{t}_{|\mathcal{T}_+|}}} \leq \frac{D_\infty^2 G_\infty \sqrt{T}}{2\alpha(1-\beta_1)}.$$

Hence,

$$\textcircled{1} = \sum_{i\in[d]} \textcircled{1}_i \leq \frac{dD_\infty^2 G_\infty \sqrt{T}}{2\alpha(1-\beta_1)}. \tag{15}$$

$$\textcircled{2} = \sum_{t\in[T]} \frac{\beta_{1,t}}{2\alpha_t(1-\beta_{1,t})}\|V_t^{1/4}(x_t-x^*)\|^2 \leq \frac{dD_\infty^2 G_\infty}{2\alpha(1-\beta_1)(1-\lambda)^2} \qquad \leftarrow \text{Lemma 4.}$$

$$\textcircled{3} = \sum_{t\in[T]} \frac{\alpha_t}{2(1-\beta_{1,t})}\|V_t^{-1/4}m_t\|^2 \leq \frac{1}{2(1-\beta_1)}\sum_{t\in[T]} \alpha_t\|V_t^{-1/4}m_t\|^2$$

$$\leq \frac{\alpha d G_\infty \sqrt{T}}{2(1-\gamma)(1-\beta_1)^2\sqrt{1-\beta_2}}. \qquad \leftarrow \text{Lemma 5}$$

$$\textcircled{4} = \sum_{t\in[T]} \frac{\alpha_t\beta_{1,t}}{2(1-\beta_{1,t})}\|V_t^{-1/4}m_{t-1}\|^2 \leq \frac{1}{2(1-\beta_1)}\sum_{t\in[T]} \alpha_t\|V_{t-1}^{-1/4}m_{t-1}\|^2$$

$$\leq \frac{1}{2(1-\beta_1)}\sum_{t\in[T]} \alpha_{t-1}\|V_{t-1}^{-1/4}m_{t-1}\|^2 = \frac{1}{2(1-\beta_1)}\sum_{t\in[T-1]} \alpha_t\|V_t^{-1/4}m_t\|^2$$

$$\leq \frac{\alpha d G_\infty \sqrt{T}}{2(1-\gamma)(1-\beta_1)^2\sqrt{1-\beta_2}}. \qquad \leftarrow \text{Lemma 5}$$

$$\text{⑤} = \sum_{t \in [T]} \frac{d\beta_1 \lambda^{t-1} D_\infty G_\infty}{1 - \beta_1} = \frac{d\beta_1 D_\infty G_\infty}{1 - \beta_1} \sum_{t \in [T]} \lambda^{t-1} \le \frac{dD_\infty G_\infty}{(1 - \beta_1)(1 - \lambda)^2}.$$

Finally, we have

$$R_T \le \frac{dD_\infty^2 G_\infty \sqrt{T}}{2\alpha(1 - \beta_1)} + \frac{d(2\alpha + D_\infty)D_\infty G_\infty}{2\alpha(1 - \beta_1)(1 - \lambda)^2} + \frac{\alpha d G_\infty \sqrt{T}}{(1 - \gamma)(1 - \beta_1)^2 \sqrt{1 - \beta_2}}.$$

$\square$

## B  Hyperparameters

### B.1  Numerical Experiment

**Distribution Shift**    For the distribution shift experiments, we used the following hyperparameters: a cycle length of 40, a learning rate $\alpha = 0.5$, exponential decay rates for the first and second moment estimates $\beta_1 = 0.9$ and $\beta_2 = 0.999$ respectively, and a small constant $\epsilon = 1 \times 10^{-8}$ to prevent division by zero. The number of time steps was set to $T = 100$.

**Noisy Samples**    For the noisy samples experiments, the hyperparameters were set as follows: a learning rate of 0.1, $\beta_1 = 0.9$, $\beta_2 = 0.999$, $\epsilon = 1 \times 10^{-8}$, and a maximum number of iterations $T = 1500$.

### B.2  CNN on Image Classification

For the CNN-based image classification experiments on the CIFAR-10 dataset, we used a learning rate of $3 \times 10^{-4}$, $\beta_1 = 0.9$, $\beta_2 = 0.999$, weight decay of 0.0005, and $\epsilon = 1 \times 10^{-8}$.

### B.3  Public Advertisement Dataset

Due to resource limitations, we performed a grid search over the learning rates for each optimizer and model using the following range: $\{\text{lr\_default}/5, \text{lr\_default}/2, \text{lr\_default}, 2 \times \text{lr\_default}, 5 \times \text{lr\_default}\}$, where lr_default is the default learning rate specified in the FuxiCTR library. We reported the best performance for each optimizer based on this search. All other hyperparameters were kept the same as those in the FuxiCTR library (Zhu et al., 2021; 2022).

## C  Additional Experiments

### C.1  Numerical Experiments

Figure 5 illustrate how both optimizers perform in a noise-free environment.

### C.2  Experiment on Resnet and Densenet

We perform experiments on Resnet(He et al., 2016) and Densenet(Huang et al., 2017) to further illustrate the effectiveness of CAdam.

### C.3  Relationship between Learning Rate, Performance, and Alignment Ratio

We tested different learning rates on the Criteo x4 001 dataset using the DeepFM model to understand the relationship between the learning rate, performance, and alignment ratio. The results in 4 show that the performance initially increases with the learning rate but starts to decline as the learning rate continues to rise. Conversely, the consistent ratio $R$ steadily decreases as the learning rate increases.

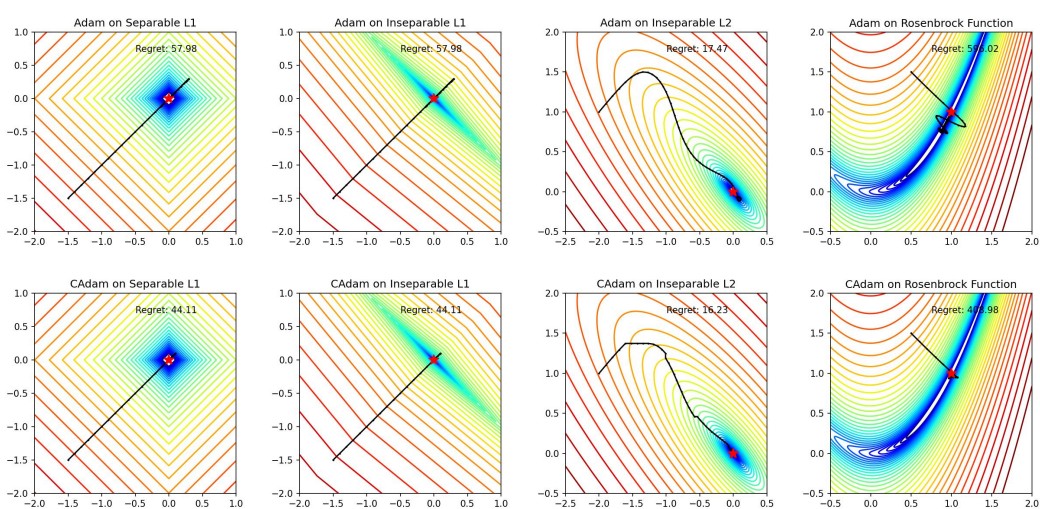

Figure 5: Performance of Adam (top row) and CAdam (bottom row) on four different optimization landscapes without noise: (Left to Right) separable L1 loss, inseparable L1 loss, inseparable L2 loss, and Rosenbrock function. This comparison highlights the natural behavior of both optimizers in a noise-free environment.

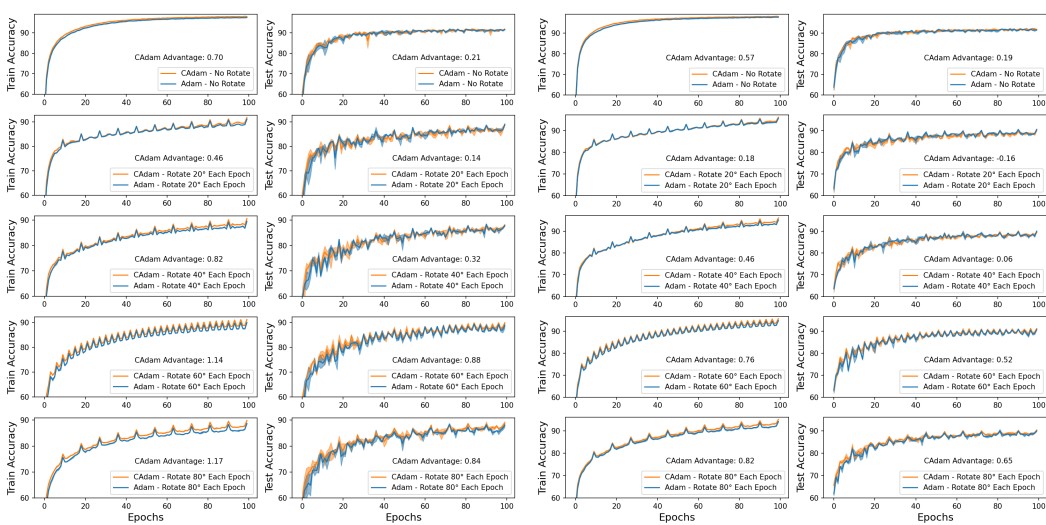

Figure 6: Performance of CAdam and Adam under different rotation speeds corresponding to sudden distribution shift. The results for Resnet are shown on the left, while those for Densenet are presented on the right.

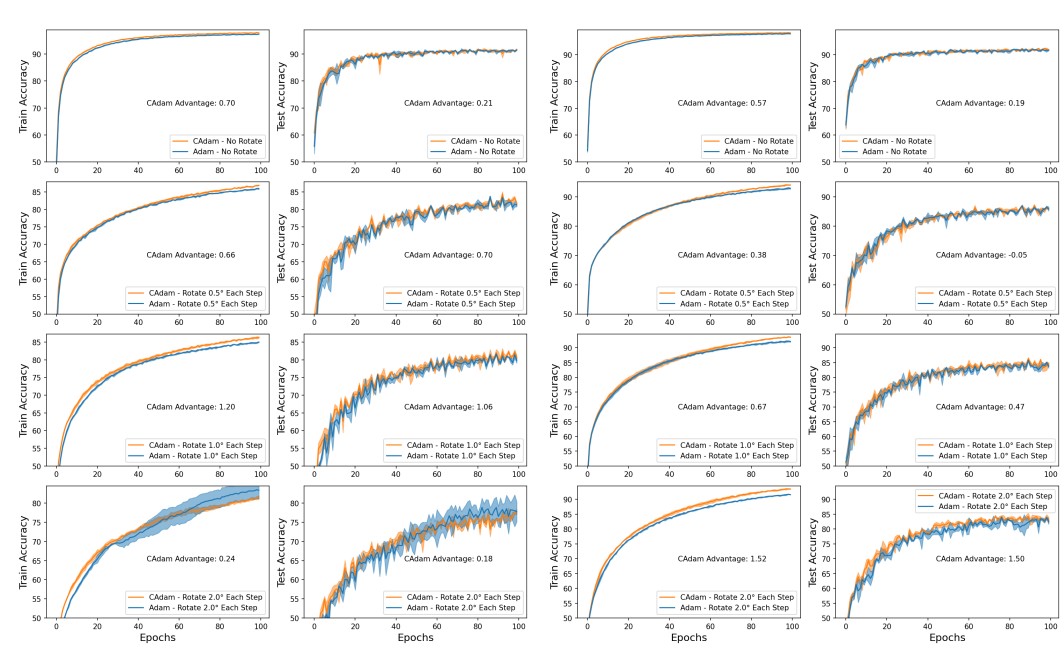

Figure 7: Performance of CAdam and Adam under different rotation speeds corresponding to continuous distribution shift. The results for Resnet are shown on the left, while those for Densenet are presented on the right.

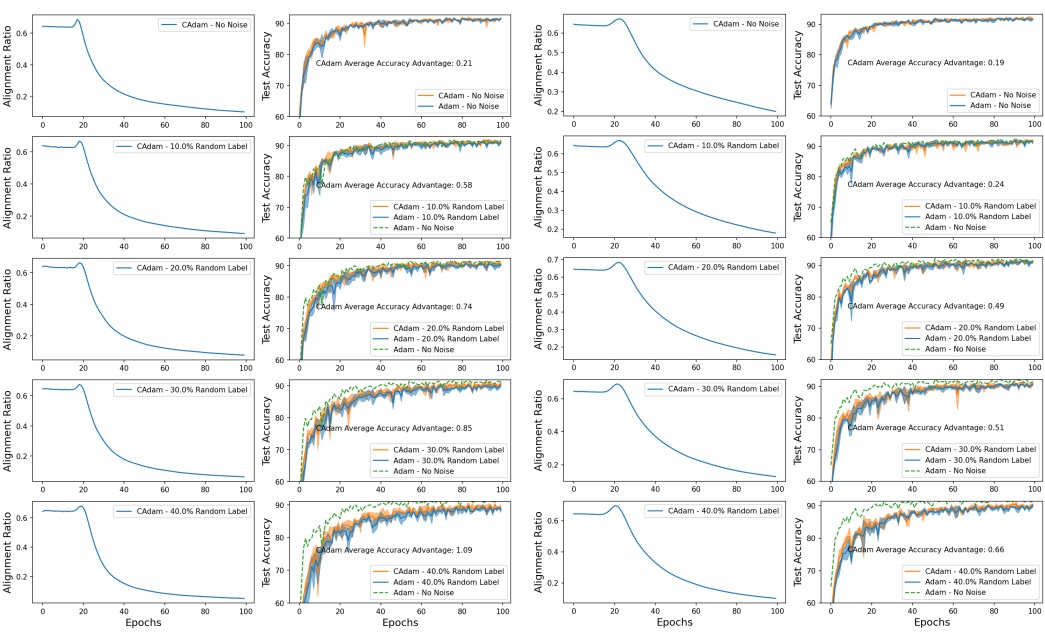

Figure 8: Performance of CAdam and Adam under noisy data. The results for Resnet are shown on the left, while those for Densenet are presented on the right.

| Learning Rate | AUC | Alignment Ratio ($R$) |
|---|---|---|
| 0.0001 | 80.59% | 63.02% |
| 0.0003 | 80.77% | 59.17% |
| 0.0005 | 80.80% | 55.78% |
| 0.001 | 80.83% | 46.45% |
| 0.0015 | 80.83% | 42.01% |
| 0.002 | 80.75% | 42.53% |
| 0.0025 | 80.66% | 41.28% |
| 0.003 | 80.55% | 37.97% |
| 0.0035 | 80.46% | 32.09% |
| 0.004 | 80.28% | 32.06% |

Table 4: Performance Metrics and Alignment Ratio for Different Learning Rates.

