# OpenReview forum: "C-Adam: Confidence-Based Optimization for Online Learning"
_ICLR.cc/2025/Conference — ICLR 2025 Conference Withdrawn Submission_

### Official Review · Reviewer_GdXb · 2024-10-18

**Soundness:** 1
**Presentation:** 1
**Contribution:** 2
**Rating:** 3
**Confidence:** 4

**Summary:**

This paper proposes a variant of popular optimizer Adam, which cancels a specific batch updating by comparing the sign of $m_{t}$ with the sign of $g_{t}$, seeing line-14 of algorithm 1.
This method is intuitive and assumes that the $g_{t}$ is affected by noisy data and should be eliminated if the sign of $g_{t}$ differs from the sign of $m_{t}$.
Regarding the evaluation, this work conducted three types of small-scope optimization tasks, an image classification task with VGG&CIFAR10 under customized distribution shift and label noise, advertisement tasks with Criteo-x4-001 dataset under different various models and optimizers, and a real-world recommendation system task.

**Strengths:**

(1) Regarding Algorithm 1, momentum term $m_{t}$ is considered a more trustworthy signal than stochastic gradient $g_{t}$ under the distribution shifts or noisy data, which makes sense to me. Then, the alignment between those two signals potentially helps identify between the clear data samples and noisy data samples.

**Weaknesses:**

$\textbf{Weakness in method motivation}$

(1) The method is intuitive and not supported by theoretical results or experiments that verify the intuition. For instance, given a new random batch $x_{i\in B}$ and the corresponding ground truth label of clear data points and noisy data points, can you explicitly show that clear data points have the same sign with $m_{t}$ while noisy data points have a different sign with $m_{t}$.

(2) Given algorithm 1 and considering line-5 and line-14, intuitively, the $m_{t}$, i.e., the cumulative behaviors, is considered as trustworthy and is utilized to filter out untrustworthy $g_{t}$ according to the sign of the two values, which make sense somehow.

However, $g_{t}$ is sampled from a small batch of data and contains noise naturally. Can you provide more insights into how line-14 of algorithm 1 is related to the natural noise of stochastic gradient?


$\textbf{Weakness in technical contributions}$

(1) This work assumes $g_{t} = \nabla f_{t}(\theta_{t-1})$ in the Notations in Section 3. Thus, this work assumes that it is a deterministic optimization problem. Along with the convex assumption, the theoretical result built upon those strong assumptions, i.e., Theorem 1, is less useful to reflect the practical performance of the proposed method.


(2) The problem is not well defined. This work mentioned the proposed method has the potential to handle distribution shifts and noise, however, what are the mathematical definitions of the distribution shifts you mentioned? And can you characterize it more formally and connect it with real-world situations? “rotating the data distribution” is confusing.

To evaluate the robustness of optimizers to noise, well-accepted datasets such as CIFAR-10-C and CIFAR-100-C may be a better choice instead of rotating the images.


$\textbf{Weakness in evaluation}$

(1) As mentioned, the customized “distribution shifts and noise” cannot support the performance improvement of the proposed method.

(2) the small scope of the experiment cannot support the performance improvement of the proposed method. Specifically, most tasks do not apply diverse settings, such as VGG network on the CIFAR-10 for image classification, Criteo-x4-001 dataset for advertisement tasks.

(3) The experiment settings are not well presented. Please refer to the Questions.

**Questions:**

(1) Figure 4, Table 1, Table 2, and Table 3 do not report the variance. Do all those experiments run only once?

(2) Can you provide more details about how the learning rate is selected for the image classification task (3e-4) and recommendation system?

(3) Why was VGG selected as the model backbone for the image classification task? since VGG is not a common selection compared with such as ResNet18 and ViT.

(4) Since the performance improvement is quite marginal comparing the proposed method with the selected baseline Adam, Can AdamW be considered as another strong baseline?

---

> ### Author Response · Authors · 2024-12-03
>
> We greatly appreciate the insightful questions and suggestions raised by the reviewer.
>
> ### Response to Motivation
>
> As mentioned in our public comments, we did find that the experimental results did not fully align with the initial intuition presented in our motivation. We will continue to explore this issue, and we sincerely thank the reviewer for bringing this to our attention.
>
> ### Response to Technical Contribution
>
> 1. We acknowledge that some assumptions in our theoretical framework may be overly strong. However, the primary contribution of this paper is not theoretical. Moving forward, we plan to strengthen the theoretical foundation by proving the convergence of CAdam under random and non-convex conditions.
> 2. In our experiments, we used "label replacement with a certain probability" to simulate noise and "rotating images" to simulate data distribution shifts. It is important to emphasize that our method is primarily designed for online learning scenarios, where training data may contain noise or experience distribution shifts. The CIFAR-C dataset is mainly used to assess the robustness of a model to noise during testing, which differs from the goals of our approach.
>
> ### Response to Evaluation
>
> **Q1:** We have re-run the experiments using three random seeds and updated the results in the paper.
>
> **Q2:** The learning rates for image classification were selected from (5e-5, 1e-4, 3e-4, 5e-4) based on Adam's performance under noise-free and rotation-free conditions. For the recommendation system experiments, the learning rates followed the default parameters recommended in the FuxiCTR repository.
>
> **Q3:** We also re-ran the experiments using ResNet and DenseNet. Due to time constraints, we will complete experiments with ViT in the future.
>
> **Q4:** We have added AdamW for comparison in our new version of paper. As noted in public comments, we believe that our improvements are not marginal.

---

### Official Review · Reviewer_DpTz · 2024-10-27

**Soundness:** 2
**Presentation:** 4
**Contribution:** 3
**Rating:** 6
**Confidence:** 3

**Summary:**

This paper introduces Confidence Adaptive Moment Estimation (CAdam). It is a variant of the frequently used Adam optimizer, in which the model parameter update of a given weight is only applied if the current gradient g has the same sign as its exponential moving average m. If this is not the case, only the exponential moving averages m and v are updated. The purpose of this change is to improve the results in online learning if distribution changes occur and/or noisy samples are present. The method is evaluated in various scenarios.

**Strengths:**

- The idea of the CAdam optimizer is simple but clever. It is easy to integrate also in many other optimizers. Hence, the proposed change can be important in the end for a broad community.
- The paper is well written and structured. The experiments are well chosen to underline the advantages of the proposed method.

**Weaknesses:**

Major:
- Is the analysis of each experiment based on a single optimization trajectory (especially those in Sections 4.1 and 4.2)? For a reliable and convincing benchmark, multiple optimization trajectories should be started from different random initializations of the model parameters. Maybe I overread this aspect. However, then this should be highlighted more. You can also employ standard deviations of the trajectories to proof that CAdam is really consistently better than Adam.
- The differences in Table 1 between CAdam and Adam are really minor. For WideDeep, the results are even equal for the given rounded numbers. But still, the CAdam results are highlighted as best and they are promoted to be consistently better. I think you are overselling CAdam here and should adjust the interpretation of your results. (CAdam still shows better performance for noisy online learning tasks, which is an important contribution by itself.)

Figure 3:
- It is difficult to see the difference between Adam and CAdam. However, there is not much insight if the accuracy is below ~50%. Maybe one could just zoom in the region between 50 and 100%. In addition, there is much white space in the three plots on the right which can be reduced.

Figure 4:
- See comment on figure 3. Try to highlight the differences of the graphs by zooming in.

Text:
- The references to Figure 3 and 4 are incorrect.

Typos:
- page 4: "optimumm"
- page 7: "corrosbonding"

**Questions:**

1. Does CAdam have a higher possibility to be trapped in local minima, since the optimization trajectory is stopped as soon as the loss would increase in a step? How does it perform for functions with more than one minimum compared to Adam averaging over multiple runs starting with different model parameter initialization?

2. Do the plateaus in the optimization process make it more difficult to identify converged results? I am aware that this issue is less severe for online learning, but there might also be many tasks where you would like to pause online learning because the training data stream is not constant.

---

> ### Author Response · Authors · 2024-12-03
>
> We appreciate the reviewer’s detailed suggestions and have made the following improvements to the paper:
>
> **W1: Random Initialization and Seeds**: For both image and recommendation experiments, we have rerun experiments with three different random seeds, and the updated results are included in the paper. For the numerical study, we agree that different initializations and hyperparameters may influence the results. Therefore, we conducted additional experiments, including:
>
> - For distribution shift experiments, we tested three cycle lengths (20, 40, 80) across various settings and starting points (-1, -0.5, 0, 0.5, 1). Both Adam and CAdam were evaluated with their respective optimal learning rates. The results are as follows:
>
>   | (setting, cycle)      | adam_regret | cadam_regret |
>   | :-------------------- | ----------: | -----------: |
>   | ('L1-linear', 20)     |     18.7589 |      12.1849 |
>   | ('L1-linear', 40)     |     9.31972 |      5.99408 |
>   | ('L1-linear', 80)     |     5.90102 |      3.78001 |
>   | ('L1-sinusoidal', 20) |     35.2794 |      23.8103 |
>   | ('L1-sinusoidal', 40) |     10.9573 |      8.55099 |
>   | ('L1-sinusoidal', 80) |     6.55878 |      4.53623 |
>   | ('L1-sudden', 20)     |     41.0581 |      25.4074 |
>   | ('L1-sudden', 40)     |     27.6424 |      16.6617 |
>   | ('L1-sudden', 80)     |      16.019 |       9.9965 |
>   | ('L2-linear', 20)     |      3.4765 |      1.78173 |
>   | ('L2-linear', 40)     |     1.66676 |     0.399042 |
>   | ('L2-linear', 80)     |     1.28224 |     0.124398 |
>   | ('L2-sinusoidal', 20) |     11.2186 |      8.11756 |
>   | ('L2-sinusoidal', 40) |     2.12727 |      1.12646 |
>   | ('L2-sinusoidal', 80) |     1.30945 |     0.333187 |
>   | ('L2-sudden', 20)     |       14.38 |      7.93337 |
>   | ('L2-sudden', 40)     |     11.0127 |      2.58427 |
>   | ('L2-sudden', 80)     |     8.16867 |      1.02211 |
>
> - For noise experiments, we tested three noise levels (0, 0.25, 0.5) and evaluated eight points within the chosen range for each optimizer, using their optimal learning rates. The results are as follows:
>
>   | (Setting, noise level)        | regret_cadam | regret_adam |
>   | :---------------------------- | -----------: | ----------: |
>   | ('Inseparable L1', 0.0)       |      28.2916 |     36.7535 |
>   | ('Inseparable L1', 0.25)      |      52.0149 |     57.2908 |
>   | ('Inseparable L1', 0.5)       |      79.5591 |     85.7855 |
>   | ('Inseparable L2', 0.0)       |      3.88095 |     6.14203 |
>   | ('Inseparable L2', 0.25)      |      5.24621 |     8.27131 |
>   | ('Inseparable L2', 0.5)       |      11.5874 |     13.9804 |
>   | ('Rosenbrock Function', 0.0)  |      253.981 |     510.723 |
>   | ('Rosenbrock Function', 0.25) |      365.139 |     633.366 |
>   | ('Rosenbrock Function', 0.5)  |      557.734 |     1058.18 |
>   | ('Separable L1', 0.0)         |      28.4953 |     35.6991 |
>   | ('Separable L1', 0.25)        |      43.9564 |     59.5631 |
>   | ('Separable L1', 0.5)         |      74.7567 |     91.0541 |
>
>
> **W2:**  We believe the improvement brought by CAdam is not marginal as noted in the public comments.
>
> > On the Criteo dataset, we observed a 0.05% improvement compared to Adam/AdamW. This improvement may seem marginal, but the maximum difference between different models on CAdam was only 0.09%. Given the already high accuracy, we argue that an accuracy improvement of the same order of magnitude as model architecture changes is not marginal.
>
> **Q1:** We believe that in the one-dimensional case, CAdam is indeed more prone to falling into local minima compared to Adam. However, in high-dimensional cases, when some dimensions are masked out, the model often "escapes" from other dimensions, making it less likely to get stuck in local minima. We conducted experiments using multiple initialization points on the Beale function (without noise), where the regret for Adam was 2906.61, and the regret for CAdam was 2417.31.
>
> **Q2:** Similarly, in high-dimensional cases, especially in deep learning scenarios, we have not observed plateauing before convergence, since at least a small number of dimensions are being updated, allowing the loss to continuously decrease.

---

### Official Review · Reviewer_3bzw · 2024-10-29

**Soundness:** 2
**Presentation:** 2
**Contribution:** 3
**Rating:** 5
**Confidence:** 2

**Summary:**

The authors of this paper propose a simple heuristic modification to the Adam algorithm to improve its performance when there is a shifting data distribution or noisy data — two common challenges in online learning for ad models/recommendation systems. The proposed algorithm, CAdam, checks if each coordinate of the gradient and the exponential moving average over the gradient share the same sign, and if not, stops updating the parameters until they align again. They also present a regret bound, seemingly closely following previous work on Adam. The authors evaluate CAdam on synthetic and real world data, including a live recommendation system, comparing against other common optimizers.

**Strengths:**

The core strength of the paper is the simplicity of the proposed modification to Adam, making it easy to deploy and reason about. Moreover, the problem of online learning under data distribution shifts and noisy data is a common problem in many real-world applications. If this simple modification can significantly improve performance under these conditions this would likely be of interest for the community.

The evaluation with A/B testing in a live recommendation system is also a promising way of demonstrating the practical benefit of the method, even if these results are not reproducible.

**Weaknesses:**

I’m leaning towards recommending rejecting the paper in its current form because of concerns about the experiments and the presentation.

My main concerns and questions regarding the experiments are:
1. I assume that in Table 1 you only show results for a single seed? Since the difference between the proposed methods and the baselines are very small, it seems crucial to make sure that the difference is not just due to random variation. I suggest running the experiment for multiple (e.g. 5) random seeds and also adding standard errors to the table. Also, when two numbers are exactly equal, you seem to only print the one corresponding to your methods in bold (e.g., WideDeep column, AMSGrad and CAdam). How do you decide what to print in bold?
2. Similarly, it would be good to show runs with multiple random seeds for the CNN image classification experiment (Figure 3 + 4).
3. Regarding the real world recommendation system results, I find it a bit hard to put them into perspective. While the scale of the experiment is impressive, it is hard to evaluate how significant the performance gains are. Is there a way to put them into context, e.g. by comparing the gains to other previous interventions? In any case, at least the gains seem to be consistent.

My main concern regarding the presentation is that despite the apparent simplicity of the algorithmic modification, its description is surprisingly unclear: in Algorithm 1, all quantities seem to be vectors, so the modification in line 14 implies that a dot product between $g_t$ and $\hat{m}_t$ is computed and, if it is smaller or equal to zero, $\hat{m}_t$ is set to the zero vector. However, the verbal description is talking about pausing the update for a single _parameter_. Also, equation (6) in appendix A is clearly stating that the comparison is applied coordinate-wise, effectively checking whether the sign of each coordinate of $g_t$ and $\hat{m}_t$  is the same. Could you please clarify this and make the description consistent throughout the paper?

If I can be convinced that CAdam is indeed a very simple way to improve performance in online learning tasks such as CTR prediction and recommendation systems and the presentation is improved, I’m happy to increase my score.

**Minor comments**

- The figures could be improved by increasing the font size and avoiding covering the letters with plot lines.
- You could consider condensing the paper a bit more, avoiding repetitive content, and removing sentences with little information.

**Questions:**

See the previous section. To reiterate the main questions:

1. Can you provide results for multiple random seeds for the experiments in Figure 3+4 and Table 1+2?
2. Can you try to put the results for the live recommendation system into perspective?
3. Can you clarify the algorithmic modification you make?

---

> ### Author Response · Authors · 2024-12-03
>
> We sincerely thank the reviewer for their constructive comments and suggestions, which have greatly inspired us.
>
> 1. **Reproducibility**: We have rerun all experiments using three random seeds and updated the results in the paper. The best-performing results are highlighted in bold, and we have retained only two decimal places in the table due to space constraints.
> 2. **Performance Improvements**: As mentioned in the official comments, even marginal performance improvements in real-world A/B tests can lead to substantial benefits. For example, a 0.1% improvement in GAUC could translate into an additional $15 million in annual revenue. Common approaches, such as feature engineering and scaling up model size, are significantly more resource-intensive than CAdam, which achieves comparable improvements at almost no additional cost. This is why we argue that CAdam’s improvements are far from marginal.
> 3. **Algorithm Clarification**: We apologize for any confusion caused earlier. The key modification in our algorithm involves masking out elements where $m_t^i \cdot g_t^i \leq 0$, i.e. $\hat{m}_t \gets \hat{m}_t \odot \mathbb{I}(m_t \odot g_t > 0)$

---

### Official Review · Reviewer_rwRk · 2024-11-02

**Soundness:** 2
**Presentation:** 2
**Contribution:** 1
**Rating:** 3
**Confidence:** 4

**Summary:**

This paper proposes a simple heuristic modification of Adam to enhance its ability to handle distribution shifts and sample noise. The modification is skipping the update of Adam if the inner product of the momentum and the gradient is negative. The performance of the algorithm is theoretically analyzed based on (Reddi et al 2018). Experiments demonstrate better performance compared to Adam and other baselines.

**Strengths:**

The writing of the paper is clear, but besides that, to be very honest, I don't think any aspect of this paper is strong by the iclr standard.

**Weaknesses:**

This paper is in my opinion another quite straightforward hack on Adam. Since Adam was proposed ten years ago there have been so many hacks on it, but it's hard to say how much they really moved the field forward. I can see at least two issues behind this, which this paper also suffers from.

- As Adam itself is heuristic, it is natural for someone to come up with many heuristic modifications based on intuitions. But do these intuitions actually reflect what's going on in the deep learning practice? No one really knows, which makes the foundation of these works quite shaky.
- It's also not hard in general to cook up some settings where the proposed hacks can help, but do they *always* help? Answering this requires very comprehensive testing which the hacking papers typically lack.

An acceptable paper of this type needs to stand out in at least one of the two dimensions.

Regarding this particular paper, I would say the proposed hack is not surprising given the intuition the authors stated ("it's bad to have momentum and gradient pointing to different directions"), but I'm not convinced of this intuition, especially due to the stochastic nonconvex nature of deep learning optimization. The experiment settings are somewhat artificial, and the actual performance gain in the experiments is marginal. I'm also not convinced that the theoretical analysis adds sufficient value to the paper, as it doesn't show how the proposed hack *improves* Adam, let alone the known limitations of (Reddi et al 2018) itself which the paper builds on.

I'm not completely denying the value of this paper, as some readers may still find it useful for their applications. But for a "competitive" conference like iclr the paper is quite far from enough.

**Questions:**

I don't have any question.

---

> ### Author Response · Authors · 2024-12-03
>
> We sincerely appreciate the reviewer’s valuable feedback and insightful suggestions.
>
> > As Adam itself is heuristic, it is natural for someone to come up with many heuristic modifications based on intuitions. But do these intuitions actually reflect what's going on in the deep learning practice? No one really knows, which makes the foundation of these works quite shaky.
>
> We acknowledge that the theoretical assumptions we build upon (e.g., from Reddi et al., 2018) are indeed strong (e.g., convexity, deterministic settings) and may not fully align with real-world deep learning scenarios. However, recent work on theoretical interpretations of Adam, such as [1], [2], and [3], has provided a more solid foundation for its use. Moving forward, we plan to incorporate these findings into our theoretical analysis to better align our work with practical deep learning applications.
>
> > It's also not hard in general to cook up some settings where the proposed hacks can help, but do they *always* help? Answering this requires very comprehensive testing which the hacking papers typically lack.
>
> We agree with the reviewer’s recommendation to conduct experiments across more diverse scenarios to better validate the generalizability of our method. That said, we would like to emphasize that our method is primarily tailored for online learning scenarios, particularly in recommendation systems, rather than as a universal algorithm for all use cases. The advertising experiments—both open-source and real-world online tests—are derived from actual application scenarios that inherently involve noise and distribution shifts. Additionally, our numerical studies and image experiments were designed as controlled, toy settings to isolate and study the distinct effects of noise and distribution shifts separately. We believe these experimental designs are natural and appropriate given the objectives of our work.
>
> [1] Zhang, Yushun, et al. "Why transformers need adam: A hessian perspective." *arXiv preprint arXiv:2402.16788* (2024)
>
> [2] Wang, Bohan, et al. "Provable adaptivity of adam under non-uniform smoothness." *Proceedings of the 30th ACM SIGKDD Conference on Knowledge Discovery and Data Mining*. 2024.
>
> [3] Ahn, Kwangjun, et al. "Understanding Adam optimizer via online learning of updates: Adam is FTRL in disguise." *arXiv preprint arXiv:2402.01567* (2024).

---

### Author Response · Authors · 2024-12-03
**Thank You and General Response to All Reviewers**

We would like to sincerely thank all reviewers for their thoughtful comments and valuable feedback. These inputs have provided us with significant insights into how the paper can be improved.

After thorough consideration, we have decided to withdraw the paper and plan for a comprehensive rewrite before resubmission. During the rebuttal process, we made substantial new experimental findings that revealed potential opportunities to further improve the algorithm. While these insights are promising, we were unable to fully address them within the limited timeframe available, and we believe a more comprehensive investigation is necessary to do justice to these developments. This has led to our decision to withdraw the paper for further refinement and improvement.

Out of respect for the reviewers, we have chosen to respond to their questions in detail and have updated the paper to address the remaining points raised.

A common concern raised by several reviewers pertains to the seemingly marginal improvements in advertising scenarios, such as the 0.05% gain over Adam/AdamW on the Criteo dataset. While this gain might appear small, it is noteworthy that the maximum performance difference between different models tested with CAdam is only 0.09%. Given the already high baseline accuracy, we argue that achieving gains comparable to those achieved through structural model improvements is far from marginal. Moreover, in real-world A/B testing scenarios involving more than 30 million active users, even minimal improvements can yield substantial benefits—for instance, a 0.1% improvement in GAUC can result in an additional 15 million dollars in annual revenue. In practical production environments, improving recommendation model performance typically relies on feature engineering or scaling up model size. The former often requires 3–5 weeks of engineering work to achieve a GAUC improvement of 0.05%, while internal experiments have shown that increasing model size tenfold improves GAUC by only 0.1%–0.3% across various scenarios, both of which are resource-intensive. In contrast, CAdam is a low-cost, vertical optimization method that delivers an average GAUC improvement of 0.3% across diverse settings, making it a viable choice for production deployment. We will enhance the discussion of this point in the next version of the paper.

Below, we address the specific points raised by each reviewer.

---

### Note · Authors · 2025-01-02

I have read and agree with the venue's withdrawal policy on behalf of myself and my co-authors.